# Prevalence, associated factors and perspectives of HIV testing among men in Uganda

**Joanita Nangendo**[1]*, **Anne R. Katahoire**[2], **Mari Armstrong-Hough**[3,4], **Jane Kabami**[1,5], **Gloria Odei Obeng-Amoako**[1], **Mercy Muwema**[1], **Fred C. Semitala**[6,7], **Charles A. Karamagi**[1,8], **Rhoda K. Wanyenze**[9], **Moses R. Kamya**[5,6], **Joan N. Kalyango**[1,10]

1 Clinical Epidemiology Unit, College of Health Sciences, Makerere University, Kampala, Uganda, 2 Child Health and Development Centre, College of Health Sciences, Makerere University, Kampala, Uganda, 3 Departments of Social and Behavioral Sciences, and Epidemiology, School of Global Public Health, New York University, New York, NY, United States of America, 4 Uganda Tuberculosis Implementation Research Consortium, Kampala, Uganda, 5 Infectious Diseases Research Collaboration, Kampala, Uganda, 6 Department of Internal Medicine, College of Health Sciences, Makerere University, Kampala, Uganda, 7 Makerere University Joint AIDS Program (MJAP), Kampala, Uganda, 8 Department of Pediatrics and Child Health, College of Health Sciences, Makerere University, Kampala, Uganda, 9 School of Public Health, College of Health Sciences, Makerere University, Kampala, Uganda, 10 Department of Pharmacy, College of Health Sciences, Makerere University, Kampala, Uganda

* joannangendo@gmail.com

**Data Availability Statement:** All relevant data are within the manuscript and its Supporting Information files.

## Abstract

### Background

Despite overall increase in HIV testing, more men than women remain untested. In 2018, 92% of Ugandan women but only 67% of men had tested for HIV. Understanding men's needs and concerns for testing could guide delivery of HIV testing services (HTS) to them. We assessed the prevalence of testing, associated factors and men's perspectives on HIV testing in urban and peri-urban communities in Central Uganda.

### Methods and findings

We conducted a parallel-convergent mixed-methods study among men in Kampala and Mpigi districts from August to September 2018. Using two-stage sampling, we selected 1340 men from Mpigi. We administered a structured questionnaire to collect data on HIV testing history, socio-demographics, self-reported HIV risk-related behaviors, barriers and facilitators to HIV testing. We also conducted 10 focus-groups with men from both districts to learn their perspectives on HIV testing. We used modified Poisson regression to assess factors associated with HIV testing and inductive thematic analysis to identify barriers and facilitators.

Though 84.0% of men reported having tested for HIV, only 65.7% had tested in the past 12-months despite nearly all (96.7%) engaging in at least one HIV risk-related behavior. Men were more likely to have tested if aged 25–49 years, Catholic, with secondary or higher education and circumcised. Being married was associated with ever-testing while being widowed or divorced was associated with testing in past 12-months. Men who engaged in HIV

**Funding:** Research reported in this publication was supported by the Fogarty International Center of the National Institutes of Health under Award Number D43 TW010037. The content is solely the responsibility of the authors and does not necessarily represent the official views of the National Institutes of Health. JN received the funding. The funders had no role in study design, data collection and analysis, decision to publish, or preparation of the manuscript.

**Competing interests:** The authors have declared that no competing interests exist.

risk-related behavior were less likely to have tested in the past 12-months. Qualitative findings showed that men varied in their perspectives about the need for testing, access to HTS and were uncertain of HIV testing and its outcomes.

## Conclusions

Recent HIV testing among men remains low. Modifying testing strategies to attract men in all age groups could improve testing uptake, reduce gender disparity and initiate risk reduction interventions.

## Introduction

HIV testing is a key entry to HIV treatment, care and prevention. Expanding coverage and access to HIV testing services (HTS) to populations with low testing rates may accelerate the HIV epidemic control and elimination efforts. Despite having over 70% of the global burden of HIV, nearly 30% of the adult population in Sub-Saharan Africa (SSA) remains unaware of their sero-status [1, 2]. By the end of 2018, only 67% of men had tested compared to 92% of women [1, 3].

Men face different challenges with HIV testing and care services compared to women. For instance, although men portray low perceived risk of HIV, they fear receiving results suggesting a positive HIV status and experience higher stigma and discrimination if diagnosed HIV + compared to women [4]. In other reports, men found facility-based services as largely feminine and contrary to valued societal norms and masculinity constructions on health service utilization [5–7]. In Uganda, however, clear understanding of men's needs to optimize engagement in HIV prevention and care remains a challenge.

An earlier study conducted among men in western Uganda showed that men's likelihood of testing for HIV varied depending on age, occupation and intention to disclose HIV status to a sexual partner [8]. More recently a study among older Ugandans aged 50+ years over half of whom were men showed that recent HIV testing (in the last 12 months) was at 53% and associated with age, self-reported sexually transmitted infections, need for circumcision and engaging in sexual activity [9]. In another study assessing preferences for uptake of community-based testing among men in Uganda access to antiretroviral therapy, availability of multi-disease testing and the mode of service delivery were reported as strongest drivers for HIV testing services [10].

Use of innovative strategies to extend reach of HTS to men could broaden opportunities through which men can learn their HIV status sero-status to address existing gender disparity in HTS delivery [1, 5]. The World Health Organisation (WHO) encourages research to better understand male-gender disparities in HIV prevention and care [3, 11]. We assessed the prevalence, associated factors and perspectives of HIV testing among men in Central Uganda.

## Methods

### Ethical considerations

We obtained ethical approval from the Higher Degrees Ethics Committee at the School of Medicine in the College of Health Sciences, Makerere University (REC REF# 2017–136) and the Uganda National Council for Science and Technology (HS226ES). We also obtained administrative permission from the administrative office of Wandegeya market, the District

Health Officer of Mpigi Health Sub-district and the local area authorities of the sampled villages. All prospective participants gave written informed consent to join the study and once enrolled were given unique codes to de-identify them. All subsequently collected data was kept confidential on a password-protected space storage only accessed by authorized study personnel.

## Study design and setting

We designed a parallel convergent mixed methods study to collect data from August to September 2018 on the prevalence, associated factors and men's perspectives of HIV testing.

We conducted a cross-sectional survey in 30 villages in the seven sub-counties of Mpigi district and enrolled participants from sampled households in each of the villages. In parallel, we conducted focus group discussions (FGDs) with men who were willing to talk about HIV testing in purposively sampled communities of Kampala and Mpigi districts. We selected these communities because they are both located in Central Uganda; the region with the highest HIV prevalence at 7.6% above national adult prevalence of 5.7% [2]. The population in these districts are predominantly semi-literate (80%), involved in subsistence farming (59.2%) and small scale retail businesses [12, 13]. In Kampala, we held FGDs in Wandegeya, a busy suburb and business centre for residents from surrounding communities of Kampala Central business district. Wandegeya has a high influx of young people since it borders Makerere University (Uganda's largest and oldest public university). It is a popular leisure and business centre predominantly operating 24 hours. We targeted men who largely spent their time or worked in, and around Wandegeya market. In Mpigi, we held FGDs in a cosmopolitan community along the Mbarara-Masaka highway (Kayabwe) which is about 80.5 Km from Kampala (Uganda's capital city). Kayabwe brings together men from the nearby fishing, business and farming communities of Mpigi district.

Mpigi district has a well-established and facilitated health service network comprising of 41 health facilities, 21 of which offer free HIV testing and Counselling and ART services on government scheme while others are on the private not for profit (PNFP) and private for profit (PFP) schemes [12]. The district has one of the highest HIV prevalence in the country at 8.0% (2, 13). Health service delivery in Mpigi is also supported by long-standing teams comprising of over 1000 community volunteers serving as Village Health Teams (VHTs) to extend services to all community members [12]. The services they provide include; immunization, Integrated Community Case Management, treatment support for malaria, HIV and TB, community mobilization and referrals [14].

## Study participants

For the survey, an individual was eligible if they were; 1) male, 2) aged ≥15 years, 3) residing in Mpigi district for ≥ 3 months prior to study recruitment and intending to live there for ≥ 6 months, 4) with an active phone contact for future communication, and 5) willing to consent to join the study. Potential participants were however excluded if they could not speak either English or Luganda (the most common local language in the region).

For the qualitative study, an individual was eligible if; 1) a male, 2) aged ≥15 years, 3) working or spending most of one's time in the selected communities and 4) willing to talk about HIV testing. Exclusion criteria were inability to speak Luganda.

## Sampling and sample size

We estimated a minimum sample size of 870 men to address the quantitative study aims. First, for prevalence of HIV testing, we used the modified Kish Leslie formula (N = $(Z^2\alpha/2P(1-P))$/

$d^2 * DE)$ [15] and considered for 95% confidence interval $(Z^2\alpha)$. We assumed a design effect (DE) = 2, 5% tolerable random error (d), 10% non-response rate and 52% prevalence of HIV testing among men in Central Uganda (P) [16]. Second, for factors associated with HIV testing, we used the formula for proportions in two independent groups $(n = (Z1+Z2)^2\ 2P\ (1-P)/(P2-P1)^2)$ [17] and assumed 5% level of significance (Z1), 80% power (Z2), 17.3% HIV testing among men involved in subsistence farming (P1), 6.0% HIV testing among men who were not involved in subsistence farming (P2) [8] and defined P = (P1+ P2)/2. Third, we used the formula for sample size estimation in cluster sampling by Bennet et al [18] $(C = P(1-P)*D/S^2 b)$ to calculate the number of clusters (villages in Mpigi district). Again, we assumed 52% prevalence of HIV testing among men in Central Uganda (P) [19], design effect (D) = 2, 95% confidence interval (S), 10% non-response rate and 30 households per cluster (b). We computed 29 clusters so we adopted the World Health Organization 30*30 cluster sampling strategy that is used for the expanded immunization programs [18].

We used two-stage cluster sampling for the survey. Villages were our primary sampling units and households the secondary sampling units. We defined a household as a group of persons related or unrelated who live and eat together in the same house [19]. In the first stage of sampling, we used the latest update of the National Population and Housing Census for Mpigi district as the sampling frame from which we selected villages using probability proportionate-to- size. We generated cumulative frequency of the total population of Mpigi district (at the time estimated to be 251512), computed the sampling interval by dividing the total population by the desired number of villages [30]. We used simple random sampling to select the first village (from a total of 339 villages) and the subsequent ones by adding the sampling interval to the population in the listing until we selected 30 villages in total. Similarly, at the second stage of sampling, we obtained an up-to-date listing of households in each village from the local area authorities for use as a sampling frame and generated a sampling interval. To start sampling in each village, we walked to a central place like a trading centre, spun a smooth bottle on a firm flat surface and took the direction in which the bottle-head pointed. We randomly selected the first house in that direction and subsequent ones by adding the sampling interval to it until we had selected at least 30 households in each village. We enrolled all eligible participants in each sampled household. When all members of the sampled household were ineligible, we recruited the most immediate neighbouring household in the direction of the botte-head.

For the qualitative study, using the prior aspects of the eligibility criteria of being a male, aged ≥15 years, willing to talk about HIV testing and working or spending most of one's time in the selected communities, we identified the first participant and recruited him to serve as mobilizer for the first FGD. After completing the first FGD, we invited the participants to serve as mobilizers for future FGDs. The required minimum number of FGDs was set depending on saturation of data which was established when there was no new information emerging from subsequent data collection and attaining maximum variation by age, education, and residence [20].

## Data collection

We collected quantitative data using a structured questionnaire designed using Open Data Kit software and administered in face-to-face interviews. One VHT from each village led the study team to collect data from that village. The study team included the principal investigator and research assistants (RAs) who were experienced in community work and HIV counselling and had at least degree level training in social sciences. Prior to the start of data collection, VHTs and RAs were trained for one week on all study procedures to clarify their responsibilities and pilot study tools. The VHTs helped in the identification of residents in the sampled households

and consenting of prospective participants. The RAs individually administered the electronic questionnaire in a hand-held tablet to each participant to capture information on socio-demographic and behavioral characteristics, HIV testing history, and barriers and facilitators to HIV testing. We defined behavioral characteristics as HIV risk-related behavior, a composite variable comprising having multiple sexual partners in the last 6 months, inconsistent condom use, engaging in transactional sex, use of alcohol and drugs, having sex after using drugs and/ or alcohol. At the end of each day, all collected data was reviewed for completeness and uploaded into a password-protected space storage designated for the study.

For the qualitative study, our aim was to explore men's perspectives on HIV testing. We held a total of 10 FGDs, 4 in Kayabwe and 6 in Wandegeya. Each FGD had 6–8 members and the conversations were held in Luganda and on average lasted an hour. Discussions were guided by a trained moderator fluent in both English and Luganda using a pre-tested topic guide. A note taker also captured details of the FGDs including body expressions which were used to enrich coding at analysis. With permission from the participants, all conversations were audio-recorded for subsequent analysis.

## Data analysis

Prior to start of analysis, we froze all the data, backed it up in Google drive and made a copy of the subsequent analyses. We transferred the data copy to Stata 14.0 for analysis. We summarized continuous variables using medians and inter-quartile ranges, and used percentages for categorical variables. We defined the outcome (HIV testing) at two levels; 1) life time testing, and 2) testing and receiving results in the last 12 months which we measured on a binary scale as a proportion and its 95% Confidence Interval (CI) adjusted for clustering at village level. We also used survey-data-restricted Poisson regression to assess for factors associated with HIV testing and measured associations as prevalence ratios (PR) and their 95% CI. At bivariate analysis, we considered variables with P<0.20 as significant for multivariate analysis but added those reported as confounders in literature even if they were not significant in bivariate tests. At multivariate analysis, we assessed for a joint association between all the selected independent variables and the outcome, and considered those with P<0.05 as significant. We also used the chunk test to check predictors for interaction, and confounding (only in absence of interaction) by testing for a $\geq$ 10% change in the effect measure in the presence of a third variable.

Audio recordings were transcribed verbatim by an independent RA skilled in qualitative research and translated to English for analysis by another RA. The principal investigator (PI) then reviewed the translated transcripts to assess accuracy and completeness. The PI also read through the transcripts to familiarize herself with the data, imported them to Open code software and applied initial open codes for inductive thematic analysis. The themes and categories which emerged were then discussed by two senior social scientists and colleagues involved in qualitative research as part of a structured working group. The feedback was used to refine codes and identify themes and illustrative quotes.

Findings from the parallel quantitative and qualitative data were triangulated at interpretation and discussion.

## Results

### Description of the study population

We screened 1389 men and recruited 1340 who met the study eligibility criteria (Fig 1). The participants had a median age of 28 years (min 15, max 83), over half (55.5%) were Catholic, 51.0% had primary level education and 43.6% engaged in farming as the main occupation.

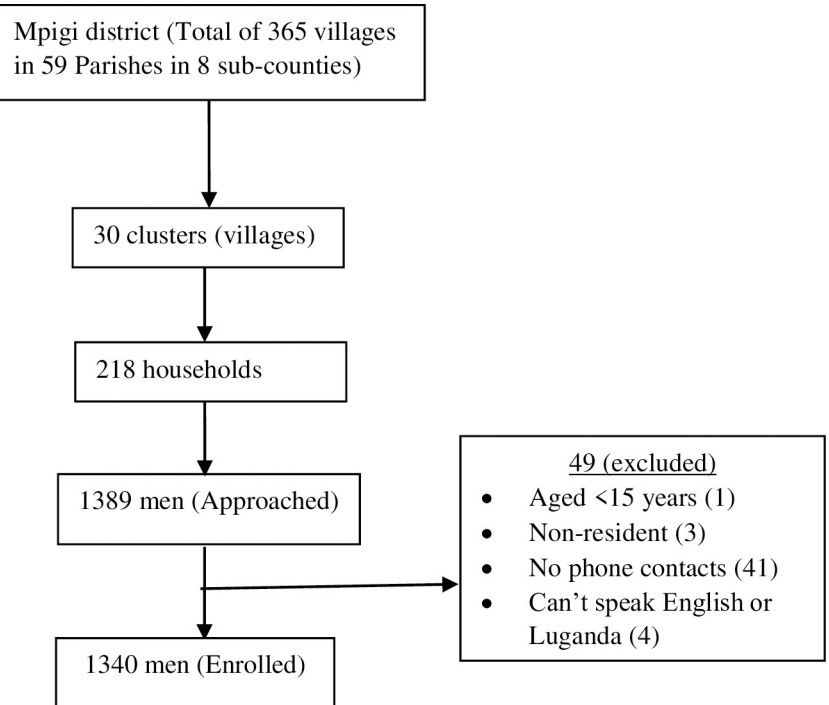

**Fig 1. Profile showing enrolment of the survey participants.**

Also, 72.5% of the men were married, 53.7% were circumcised and nearly all (96.7%) engaged in at least one HIV risk-related behavior (Table 1).

## Prevalence of HIV testing

Eighty-four percent (1126/1340) of the participants reported having had tested for HIV in their life time but only 65.7% (880/1340) reported testing in the last 12 months. The majority (91.8%) were aware of HTS within their locality, 83.4% had tested for HIV at the local HTS service point and nearly all (98.8%) who had tested in their lifetime had received their test results. Of the participants who knew their partner's HIV status (56.4%), only 67.5% had ever tested as a couple (Table 2).

## Factors associated with HIV testing

Men were more likely to have ever tested for HIV in their lifetime if they were; aged 25–49 years (aPR: 1.12; 95%CI: 1.06–1.18), Catholic (aPR: 1.10; 95%CI: 1.02–1.18) or of other religious denominations (aPR: 1.11; 95%CI: 1.01–1.22), with ordinary (aPR: 1.24; 95%CI: 1.06–1.47) or advanced secondary or higher education (aPR: 1.32; 95%CI: 1.11–1.57), married (aPR: 1.17; 95%CI: 1.09–1.26) and circumcised (aPR: 1.08; 95%CI: 1.04–1.13).

Similarly, men were more likely to have tested for HIV in the last 12 months if they were; aged 25–49 years (aPR: 1.14; 95%CI: 1.06–1.22), Catholic (aPR: 1.14; 95%CI: 1.05–1.25) or of other religious denominations (aPR: 1.19; 95%CI: 1.06–1.34), with ordinary (aPR: 1.24; 95% CI: 1.04–1.47) or advanced secondary or higher education (aPR: 1.28; 95%CI: 1.07–1.53), widowed or divorced (aPR: 1.58; 95%CI: 1.09–2.31), circumcised (aPR: 1.16; 95%CI: 1.10–1.23), and with a professional occupation (aPR: 1.21; 95%CI: 1.01–1.46) or involved business/trading (aPR: 1.23; 95%CI: 1.03–1.45).

**Table 1. Socio-demographic and behavioral characteristics of survey participants from selected villages in Mpigi district, August to September 2018 (N = 1340).**

| Variable | Measure |
|---|---|
| Median age in years, (IQR) | 28 (21, 39) |
| Median household size, (min, max) | 3 (1, 12) |
| Religion, n (%) | |
| Moslem | 254 (18.9) |
| Anglican | 238 (17.8) |
| Catholic | 744 (55.5) |
| Others* | 104 (7.8) |
| Education, n (%) | |
| No formal education | 47 (3.5) |
| Primary (P1-P7) | 683 (51.0) |
| Ordinary secondary (S1-S4) | 458 (34.2) |
| Advanced secondary (S5-S6) & higher | 152 (11.3) |
| Occupation, n (%) | |
| Farming | 584 (43.6) |
| Professional[a] | 244 (18.2) |
| Business/trading | 377 (28.1) |
| Unemployed | 88 (6.6) |
| Students | 28 (2.1) |
| Others[#] | 19 (1.4) |
| Marital status, n (%) | |
| Single | 315 (23.5) |
| Married | 971 (72.5) |
| Widowed, divorced, separated | 54 (4.0) |
| Being circumcised (Yes), n (%) | 719 (53.7) |
| Engaging in HIV risk-related behavior[+] (Yes), n (%) | 1296 (96.7) |

* Pentecostal (75/1340), Seventh day Adventist (27/1340), African traditionalist (1/1340) & Isamasiya (1/1340)

[a] Professional those practicing in a line of work after attaining formal qualification

[#] casual laborer (10/19), Parish chief (1/19), school cook (1/19), Preacher (1/19), Musician (1/19), Traditional healer (1/19), Lumber (4/19), [+] HIV risk-related behavior include having multiple sexual partners in the last 6 months, inconsistent condom use, engaging in transactional sex, use of alcohol and drugs, having sex after using drugs and alcohol as in S1 Table

There was interaction between marital status and engaging in HIV risk-related behavior. Those who were either widowed, divorced or separated but involved in risk-related behavior had 39% reduction in likelihood to have tested for HIV in the past 12 months compared to those who were either single, widowed and divorced but not involved in any risk-related behavior (aPR: 0.61; 95%CI: 0.40–0.92). Occupation and HIV related-risk behavior were not significant at bivariate analysis but were included in the multivariate analysis because they were known confounders in literature (Table 3).

## Barriers, facilitators and views of how to improve HTS

**Quantitative findings.** Men reported several barriers to HIV testing including perceiving themselves as HIV negative with no need to test (47.2%), fear of learning their HIV status (24.7%), mere lack of interest in testing (20.5%), having busy work schedules (17.6%), mistrust of HIV testing methods (17.6%) and fear of testing-related gossip (11.7%) (Table 4).

**Table 2. HIV testing history among male survey participants from selected villages in Mpigi district, August to September 2018 (N = 1340).**

| Variable | Measure |
|---|---|
| **Outcome variable** | |
| Ever tested in one's lifetime (Yes), n (%; 95%CI) | 1126 (84.0; 80.6–87.0) |
| Tested in last 12 months | 880 (65.7; 60.4–70.6) |
| Tested $\geq$ 12 months ago | 246 (18.4; 15.1–22.1) |
| Never tested in lifetime | 214 (15.9; 13.0–19.4) |
| **Other variables on HIV testing history** | |
| Aware of nearby HIV testing services (Yes), n (%) | 1230 (91.8) |
| Testing at service point in area of residence (Yes)[+], n (%) | 1026 (83.4) |
| Waited & received test results (Yes)[*], n (%) | 1113 (98.8) |
| Last HIV test results (HIV+), (n (%) | 57 (4.3) |
| Disclosure HIV status (Yes)[*], n (%) | 834 (75.0) |
| Partner's HIV status awareness (Yes), n (%) | 756 (56.4) |
| Ever tested with partner (Yes)[a], n (%) | 510 (67.5) |

[+]Among those aware of nearby HIV testing services (N = 1230)

[*]Among those who have ever tested for HIV (N = 1126)

[a] Among those who knew their partner's HIV status (N = 756)

Men were motivated to test if; their lifestyle involved HIV risk-related behaviors (47.7%), were encouraged by friends (37.5%), had a new partner (24.0%), and accessed free testing (23.4%). Suspecting oneself as HIV+ (44.0%) was however both a barrier and motivation for testing. Men reported they may be more motivated to test if they were taught the benefits of HIV testing (75.5%), testing services were brought closer (62.7%), free testing was more accessible (52.9%) and they were given a chance to use new HIV tests (47.3%) (Table 4).

**Men's perspectives on HIV testing.** The qualitative study explored men's perspectives of HIV testing. Several themes emerged that illuminated men's varied perspectives regarding HIV testing. The themes ranged from the belief that there is no need to test; or that testing is only for specific purposes and categories of men; to uncertainty, fear and mistrust of the testing processes and outcomes. Among the key challenges highlighted were opportunity costs as well as challenges in accessing medications after testing.

*No need to test*. Quantitative findings showed that one of the major barriers to men's HIV testing was perceiving oneself as HIV negative (47.2%). Similarly, from the qualitative findings' men thought of themselves as safe and felt no need to test whatsoever.

> "... it is true that you may go and test for HIV, but I don't see any reason to as why I should go for testing when I don't have it." (FGD2 Wandegeya)

Men felt that knowing their partner's status would suffice and that they did not have need to test. Likewise, in the quantitative findings, few men reported testing because their partner was pregnant (2.5%) but otherwise wouldn't have.

> "... for the men when their wives test and find they are negative; they instead find no need to test
>
> as they think they are also negative." (FGD3 Kayabwe)

**Table 3. Socio-demographic and behavioral characteristics associated with HIV testing among male survey participants from selected villages in Mpigi district, August to September 2018.**

| Independent variables | Ever tested for HIV in life time [a] | | Tested for HIV in last 12 months [b] | |
|---|---|---|---|---|
| | Unadjusted prevalence ratio (95%CI) | Adjusted prevalence ratio (95%CI) | Unadjusted prevalence ratio (95%CI) | Adjusted prevalence ratio (95%CI) |
| Age | | | | |
| 15–24 | 1.00 | 1.00 | 1.00 | 1.00 |
| 25–49 | 1.14 (1.09–1.19) | 1.12 (1.06–1.18) | 1.14 (1.08–1.21) | 1.14 (1.06–1.22) |
| 50–83 | 1.06 (0.95–1.17) | 1.07 (0.96–1.20) | 1.04 (0.94–1.15) | 1.09 (0.97–1.23) |
| Religion | | | | |
| Moslem | 1.00 | 1.00 | 1.00 | 1.00 |
| Anglican | 1.00 (0.91–1.10) | 1.04 (0.95–1.15) | 1.02 (0.91–1.14) | 1.10 (0.97–1.25) |
| Catholic | 1.05 (0.96–1.14) | 1.10 (1.02–1.18) | 1.05 (0.95–1.15) | 1.14 (1.05–1.25) |
| Others | 1.06 (0.95–1.18) | 1.11 (1.01–1.22) | 1.11 (0.98–1.25) | 1.19 (1.06–1.34) |
| Education | | | | |
| No formal education | 1.00 | 1.00 | 1.00 | 1.00 |
| Primary level (P1-P7) | 1.17 (1.00–1.37) | 1.17 (1.00–1.37) | 1.14 (0.96–1.35) | 1.14 (0.97–1.35) |
| O-level (S1-S4) | 1.21 (1.04–1.42) | 1.24 (1.06–1.47) | 1.20 (1.02–1.42) | 1.24 (1.04–1.47) |
| A-level & higher | 1.32 (1.12–1.56) | 1.32 (1.11–1.57) | 1.29 (1.08–1.54) | 1.28 (1.07–1.53) |
| Marital status | | | | |
| Single | 1.00 | 1.00 | 1.00 | 1.00 |
| Married | 1.23 (1.14–1.33) | 1.17 (1.09–1.26) | 1.22 (1.14–1.30) | 1.21 (089–1.65) |
| Widowed & divorced | 1.14 (0.94–1.37) | 1.11 (0.91–1.32) | 1.00 (0.82–1.23) | 1.58 (1.09–2.31) |
| Being circumcised | | | | |
| No | 1.00 | 1.00 | 1.00 | 1.00 |
| Yes | 1.04 (0.98–1.10) | 1.08 (1.04–1.13) | 1.10 (1.03–1.17) | 1.16 (1.10–1.23) |
| Occupation | | | | |
| Unemployed | 1.00 | 1.00 | 1.00 | 1.00 |
| Farming | 1.13 (0.97–1.32) | 1.09 (0.92–1.28) | 1.17 (1.00–1.34) | 1.14 (0.96–1.37) |
| Professional | 1.22 (1.02–1.46) | 1.14 (0.95–1.37) | 1.29 (1.08–1.54) | 1.21 (1.01–1.46) |
| Business/trading | 1.20 (1.04–1.40) | 1.14 (0.97–1.34) | 1.28 (1.10–1.49) | 1.23 (1.03–1.45) |
| Students & others | 1.02 (0.80–1.32) | 1.03 (0.79–1.34) | 1.10 (0.84–1.43) | 1.08 (0.82–1.44) |
| HIV risk-related behavior | | | | |
| No | 1.00 | 1.00 | 1.00 | 1.00 |
| Yes | 1.06 (0.91–1.24) | 1.03 (0.88–1.20) | 1.08 (0.89–1.31) | 1.10 (0.78–1.55) |
| Maritalstatus by HIV risk-related behavior | | | | |
| Married#Yes | | | | 0.95 (0.70–1.30) |
| Widowed&divorced#Yes | | | | 0.61 (0.40–0.92) |

*Testing is for specific purposes.* Just as a lifestyle with risk-related behaviors motivated many men (47.7%) to test in our quantitative findings, men in the FGDs reported that testing was for those who were unsure of their sexual life styles.

"... if you have loved over 4 to 5 women, you go for testing when you are scared, may be you were not sure of one's status, may be one woman has signs and you have ever had sex with her so you also expect your life not to be healthy which drives you to go for testing to know your status." (FGD2 Wandegeya)

**Table 4. Barriers, facilitators and views of how to improve HIV testing among male survey participants in Mpigi district, August to September 2018.**

| Variable | Measure |
|---|---|
| **Barriers to HIV testing**\*, n (%) | |
| **Psychosocial barriers** | |
| Perceiving self as HIV- | 656 (47.2) |
| Fear to know one's status | 343 (24.7) |
| Merely not interested | 285 (20.5) |
| Busy work schedules | 236 (17.6) |
| Fear of testing-related gossip | 162 (11.7) |
| Fear of being pricked | 112 (8.4) |
| **Health service barriers** | |
| Mistrust of testing methods | 236 (17.6) |
| Long wait for tests and results at facilities | 133 (9.9) |
| More need for privacy to test | 109 (8.1) |
| Long distance to testing places | 109 (8.1) |
| High costs for HIV testing | 98 (7.3) |
| Poor health worker attitudes | 77 (5.8) |
| Facility services are male unfriendly | 53 (4.0) |
| Early close-up times at testing centers | 49 (3.7) |
| **Facilitators of HIV testing**\*, n (%) | |
| Lifestyle involves risk-related behavior | 639 (47.7) |
| Friends encouraged me | 503 (37.5) |
| Had a new partner | 321 (24.0) |
| Free testing was nearby | 314 (23.4) |
| To know my status | 274 (19.7) |
| Knew about availability of Antiretroviral therapy | 166 (12.0) |
| Needed to be circumcised | 101 (7.5) |
| Someone close to me tested positive | 64 (6.9) |
| I can pay to test | 50 (3.7) |
| My partner was pregnant | 33 (2.5) |
| **Overlapping as a barrier and facilitator** | |
| Suspecting self to be HIV+ | 611 (44.0%) |
| **Suggestions of how to improve HIV testing**\*, n (%) | |
| More sensitization on benefits of testing | |
| Bring testing services closer | |
| Make free testing more available | |
| Bring new tests | |
| Reduce prices for testing | 365 (27.2) |

\* The responses were obtained using multiple choice questions

Findings from the quantitative study showed that men (7.5%) were motivated to test when they needed to be circumcised, in the FGDs, men indicated that testing was for those seeking medical male circumcision as a prerequisite before being operated on.

> "...I went for testing when I had gone for safe male circumcision, they said if you know that you are positive, we shall not circumcise you." (FGD2 Wandegeya)

Men said that they would test only if one got symptoms of an unexplained illness or if a partner died of a wary illness.

*"Listen, for me unless. . . I will start seeing the signs that I am now positive. . ., I cannot leave here walk straight to the health facility that I am going for HIV testing, I cannot." (FGD2 Wandegeya)*

*". . . you can know at that a man has been infected by HIV virus, or that you have the disease if you learn that your former loved one has passed on, . . . but if none of my lovers have never died, then no. "(FGD2 Wandegeya)*

Men also said that testing at times is a compulsory service for the ill at health facilities to identify their ailment.

*". . . treatment has changed, . . . if you go the health facility, they force you to test for HIV also, . . . that is how they get the chance of testing the gentlemen, but it is very difficult to find one going by himself for HIV." (FGD2 Wandegeya)*

Some men also felt that testing is easier if one is with a casual partner as they might not query the decision to test but others argued that testing makes no difference if a partner is unfaithful.

*". . .it is very hard to be sure that they abstain to wait for you up to when you will return to them since you both sleep in different places. . . but that is exceptional for one with whom you share the bed every night." (FGD1 Wandegeya)*

*"At one point you can say that we have finished with testing. . . but she will still cheat since you can't have the same steamy love all through as it was in the beginning." (FGD1 Wandegeya)*

*Fear of the testing processes and outcomes.* Some men reported that they simply disliked testing because they did not want to be pricked. Our quantitative findings showed that 8.4% of men feared injections. Similar sentiments were expressed in the FGDs.

*"According to me, the biggest reasons for not testing is the injection." (FGD3 Kayabwe)*

Our quantitative findings showed that 24.7% of men feared to know their status. Similarly, from the qualitative study some men reported that the decision to test was equivalent to signing a death sentence because testing HIV+ could end one's dreams. Others however said that HIV was no longer life-threatening because ART was readily available.

*". . . you have your plans, but now you have been informed that you are positive. How will you inform your father that I acquired HIV, when you look through all those situations, you say, please to hell with testing." (FGD1 Wandegeya)*

*". . .HIV is like as any other disease because ARVs are readily available, so that pressure you are talking about is no longer too much, it reduced seriously." (FGD5 Wandegeya)*

Unique to the qualitative study, men had the following perspectives.

*Testing is for only certain categories of men*. Some men also felt that testing was for the poor because they are easily lured into sex for money and get rich schemes which put them at higher risk of HIV infection.

> "...he is looking for a woman who has constructed, has enough money and will only give him the job of staying home, ...they want modern things, now he got a mature lady, she infected him." (FGD4 Wandegeya)

Some men said that receiving HIV positive results could easily cause one to engage in risk-related behavior and ignore HIV prevention and treatment services but others argued that HIV negative results could motivate positive behavior change.

> "... when people get to know that they are positive they don't mind putting on a condom they say I will not die alone, meaning he will not use a condom he will go on spreading to others." (FGD5 Wandegeya)

> "... if you test for HIV and God helps you and you find yourself HIV negative, you can reduce speed at which you have been moving and say ... let me be with one person." (FGD4 Kayambwe).

Our quantitative results also showed that 37.5% of men tested because they had been encouraged by friends. Similarly, from the FGDs, men said that the company of a friend could comfort and encourage one to test.

> "Sometimes you can test because your friend has tested." (FGD3 Kayabwe).

This underscores the need for encouragement and support in getting men to test

*Accessing health services and medications*. From the quantitative findings, 23.4% of men accessed and tested using free nearby services while 62.7% and 52.9% suggested bringing services closer and free testing respectively to attract more men. In the FGDs some men confessed having tested for HIV several times, while others had never tested but were willing to do so if changes could be made to the current HTS;

Men reported that testing could be easy if services were closer for example in the form of free HIV testing and integrated disease outreaches.

> "... send us health workers to come to test us and find us where we are in the villages for free." (FGD2 Kayabwe)

> "... not only testing for HIV/AIDS, there are other diseases like diabetes, hypertension and syphilis, to me, such diseases also need to be tested for because others think that HIV is the only disease." (FGD4 Wandegeya)

Again in our quantitative findings 7.3% of men cited high costs as a barrier to testing which was in-part mirrored in the FGDs when men said that testing requires one to have extra money for transport costs. However, some still preferred paying to test at private clinics while others suggested offering of occasional incentives to attract more men to test.

> "... if someone decided to test, some can say they cannot stand in tents, ... someone will say my clinic is Abbi, but to go to Abbi to test, they will request for 15,000/ = [equivalent to about 4 USD]." (FGD1 Wandegeya)

*". . . if they hear there is testing but they also offer a token of 5000/ =* [equivalent to about 1.35 USD] *that can increase the number of men turning up for testing, . . .they can come to test and get that money and later go to do other activities." (FGD1 Kayabwe)*

Upon testing, men complained of the difficulty to access ART and Post Exposure Prophylaxis (PEP) if tested HIV positive or exposed to HIV infection respectively since it involves lining up which can be both tiring and shaming.

*"Testing wouldn't be bad, but the challenge after they test you positive,. . . you have to line up. Some even fear to test and know their status because they will be seen lining up for medicines." (FGD4 Wandegeya)*

*". . . just as they quicken for those of accidents, even us who are not of accident but were infected, they should also quicken for us. I should also be given that PEP." (FGD3 Wandegeya)*

Other men however said that increased availability of ART and PEP falsely present HIV as non-life-threatening which weakens efforts to reduce HIV transmission.

*". . .there is a lot of medicines that you can get someone who is truly infected and you cannot know that he is really infected. Even though they tell you, you cannot believe it." (FGD2 Wandegeya)*

*Opportunity cost.* In both our quantitative and qualitative findings, men complained that testing requires a lot of time which they are unable to set aside regularly for several reasons including; busy work schedules (17.6%), waiting to be tested and receive results (9.9%) and testing-centre close-up times (3.7%).

*". . .it is good for men to go and test but we don't always have time to go and test, we always have a lot of work, everything demands of your time, and by the time it gets to evening, it is when you find a window. The truth is that we really fail to get time to go and test." (FGD1 Kayabwe)*

The men however suggested that increasing the number of health workers allocated to HIV testing could in-part address the delays in testing.

*". . . increase the number of health workers who come to villages. In towns like this one they come and we converge in one place and they test us. Now there is coming and you come 3, like how you found me when I'm in a hurry going; then you start wondering lining up." (FGD2 Kayabwe)*

*Uncertainty and mistrust of the testing process.* Men expressed mistrust in commonly used HIV tests over being counterfeits so the results they yield can't be trusted. Some also complained about drugs which if swallowed prior to testing could mask HIV resulting in false negative tests.

*"I have a friend of mine. . . at first he was told he is positive. When he came here, he was told he is negative. That's how I ended up not testing myself. I don't want to test and they start making errors on my life." (FGD3 Kayabwe)*

*". . . if you have found a partner and you choose to go and test together they can swallow certain drugs even if it is instant and the virus will hide that even if they test, they will still return with negative results." (FGD1 Wandegeya)*

Men were also not satisfied with the privacy and confidentiality offered in testing. They complained of gossip among health workers and social discredit from community members who find them waiting at HIV testing sites. They suggested alternating the health workers to reduce familiarity so as to improve the privacy of clients.

*"I can't fear telling you my issue because I don't know you and if you know me you will keep quiet you won't tell it to your colleagues because you don't know me." (FGD4 Kayabwe)*

*"When I go for testing, I may find someone from my village who can go and tell everyone that I am HIV positive. When everyone gets to know that you are HIV positive, they will invade your privacy." (FGD3 Kayabwe)*

Men however said that even if they tested, it would be difficult to disclose the HIV positive status to their partners because it may destabilize or end a rather peaceful relationship but felt that if tested as a couple it would be easier to accept the results and trust each other.

*"I don't tell her, however I go and get the medication, if it comes out that, maybe when she has gone for antenatal care . . . and tests and discovers that she has it (HIV). I tell her that we are all just starting." (FGD2 Kayabwe)*

*". . . when we go with my wife, she may have doubted me every day but if we happen to go and they test us for HIV, she will trust me and I will also trust her." (FGD4 Kayabwe)*

They also expressed fear of taking life-long HIV treatment if confirmed HIV+ though many were confident that if one adhered to the prescribed regimens they would live a healthy life with reduced risk of spreading the infection.

*". . . when you test and find out you are positive; you worry so much. The medicine is very bitter and yet they are to be taken daily." (FGD3 Kayabwe)*

*". . . what I know is that if a woman is taking ARVs when the man is HIV negative, when she takes ARVS everyday her infection is very hard to transmit because it is dormant." (FGD4 Kayabwe)*

Men expressed partial trust in health workers offering HIV services because they are easily bribed to issue wrong results if a requesting client makes a special payment. They also complained about health workers' poor attitudes especially to clients seeking free services in public facilities.

*". . .some health workers hide people's results and release incorrect results because they have given him/her money." (FGD4 Kayabwe)*

*". . . health workers in the health facilities are very rude, if you just go to the health facility that treatment is free, someone will look at you like you are nothing and you have to first pay something. . . . and he feels like he is just helping you. "(FGD2 Wandegeya*

## Discussion

Compared to the national standard of 95% testing in the general population, we found that while men had a moderately-high prevalence of lifetime testing at 84%, this dropped to 66% when it came to testing in the last 12 months. These findings are similar to the national HIV testing prevalence of 81% in adults and 67% in men alone [3]. Although short of the adopted UNAIDS 95-95-95 targets, Uganda attained a 25% increase in overall testing up from 56% in 2011 [3, 16]. Nevertheless, HIV testing remains lower among men than in women despite concerted efforts by MoH to extend HIV services [3]. Our findings highlight men's testing as a prevailing bottleneck in the HIV care cascade in Uganda thus underscoring the need to understand men's challenges to HIV testing, prevention and care.

We found that men were more likely to have ever tested and tested in the last 12 months if they were; aged 25–49 years, Catholic, with secondary or higher education and seeking for medical male circumcision. This agrees with earlier studies which showed that older youths and young adults aged 25–49 years were more independent and so could easily articulate individual need for health services and seek them unlike their younger and older counterparts who often need to seek consent or assistance from a significant other [6, 21, 22]. In South Africa, a study reported increased testing among men including first-time-testers when approached through religious leaders in churches [23]. Similarly, in Tanzania, religion has been shown to motivate positive behavior change including HIV testing, stigma reduction and sero-status disclosure because it provided confidence of the supremacy and over-ruling power of a higher God who is able to care for all [24]. From both studies there is clear emphasis that religion is a significant social force on lifestyle, behavior and health service utilization in SSA [23, 24]. Similarly, reports from Uganda and elsewhere in SSA affirmed that having secondary or higher education positively influences individual autonomy and cognitive ability to appreciate the benefit of testing and subsequent decision-making on healthcare services like testing and medical male circumcision [25–28]. Medical male circumcision has for long been known as a prevention measure for HIV negative men with ability to reduce the risk of HIV transmission by 60% [29]. However, prior to being offered, one is mandated to test so as to obtain sero-status results which are used to guide safety precautions taken in the surgical procedure, subsequent care and treatment [29]. This arrangement thus positions medical male circumcision as a significant driver for HIV testing among men considering to use the service [29]. Nonetheless, male-driven sensitization campaigns could be used to reach more men irrespective of their education level so as to improve their responsiveness to HIV prevention and care programs [30].

Concerning testing in the last 12 months, we found that married men were more likely to have tested compared to the unmarried. Men were however less likely to test if engaged in HIV risk-related behaviors. Our results agree with earlier studies from Uganda and Zambia which showed that married men are more likely to have multiple lifetime tests than unmarried men because of positive influence from their partners/spouses especially if they too have ever-tested. Spousal influence initiates general behavior change especially improved healthcare service utilization, health seeking patterns and more gradually the definition of rigid masculinity definitions and values [6, 21, 25, 31]. We also observed that the drop in testing at the last 12 months could be due to several reasons including prior awareness of a positive HIV status which was reflected by reduced need to access and/or engage in availed HTS. Nonetheless, through alternative HTS modalities like antenatal care testing, community-based testing, home-based testing and HIV self-testing which prioritize couple involvement, men could be encouraged to test more regularly [30, 32].

From our parallel qualitative study, several themes emerged that highlighted men's varied perspectives regarding HIV testing; these ranged from the belief that there is no need to test; or that testing is only for specific purposes and certain categories of men; to uncertainty, fear and mistrust of the testing processes and outcomes. Other emerging themes included opportunity costs as well as challenges in accessing services as well as medications after testing.

First, concerning who needs to test, men felt testing is relevant if one was ill, in need of circumcision, unsure of their sexual lifestyle, had a pregnant partner or offered as a mandatory health service. They otherwise felt no need to test because they perceived themselves as HIV negative, thought of it as a waste of time if a partner is unfaithful, equated testing to signing a death sentence and believed testing HIV+ could instead ignite reckless behavior. Our results align with those from a study in Lesotho which showed that men largely felt testing was for women because they are assumed to be responsible for bringing HIV infection in a relationship. This reflected societal reinforcement of hegemonic masculinity and gender power inequities [33, 34]. DiCarlo also found that men considered testing was a life-changing event regardless of the results which was incongruous with their lifestyles. Furthermore, men looked at testing as a 'death sentence' which they would rather avoid because of struggles with community-based stigma, fear of relationship conflicts, dearth of information on benefits of ART and positive living if diagnosed HIV+ [33]. This highlights the need for gender-sensitive behavioral interventions to address influence of stigma and masculinity constructions on uptake of HIV services in our predominantly patriarchal societies.

Second, about access to HTS, men testified being encouraged by friends to test, expressed preference for free testing, integrated disease outreaches and incentivized testing but complained of high costs of HIV tests, time delays in testing and difficulties in accessing HIV drugs (ART and PEP). These findings match those presented from a male-catchup plan in Tanzania where strategies to extend HTS to men included biomedical and non-biomedical approaches like expanding targeted HIV testing, HIV self-testing, integrating HIV and other health services, as well as socio-cultural approaches to encourage health seeking and safe behaviors and peer education programs [35]. Similarly, a recent systematic review on evidence of interventions to improve men's HTS, suggested tailoring of information on available HIV services suitable for men [30]. HIV programs could prioritize expansion and implementation of recommended male-friendly testing services like HIV self-testing, home-based testing, community-based testing and antenatal care testing.

Third, regarding uncertainty of HIV testing and its outcomes, men complained about the presence of counterfeits and poor quality HIV tests, unclear messages on HIV testing and safe male circumcision, lack of trust in health workers offering HTS, compromised privacy and confidentiality in testing, difficulties in HIV status disclosure and fear of HIV life-long treatment. The findings agree with a systematic review on facilitators and barriers of testing in Zambia which showed that misconception of HIV testing and fear of negative consequences such as stigma, discrimination and breach of privacy hindered utilization of HTS especially among men [36]. Developing and implementing interventions that address HIV misconceptions and stigma could improve men's engagement in HIV testing, prevention and care programs.

Our study met some limitations. First, we collected self-reported data of HIV testing so it is possible that some participants may have given false information of their testing status introducing measurement bias. Though people usually over-estimate positive phenomenon with self-reports, this may have been minimal since our findings remained below the set 95% national testing targets so it is likely that people largely spoke the truth about their health testing status [37, 38].

Second, we collected our data using a structured questionnaire which may have limited the possible range of responses to different aspects of HIV testing. We however minimized this by adopting measures used in reporting of national statistics for uniformity [3]. Also, the use of focus group discussions to collect data on perspectives of HIV testing may have missed individual concerns but rather extracted data reflecting social influences on testing. Nonetheless, we used a mixed methods approach to collect data which offered complementarity for inherent limitations of individual data collection methods [39]. Third, we used a cross-sectional study design which may have introduced selection bias because we only had access to people available in the study setting at the time. We however minimized the bias by using two-stage sampling strategy to select the participants [40]. We could also have introduced selection bias by excluding men who could not speak either Luganda or English but this was minimal since most of the participants encountered were well able to communicate with us in at least one of the set languages. Still, because of our choice study design, we could not entirely rule out confounding but at analysis checked for the effect of a third variable and found none. Also, since we met participants as a one-off, we could not account for the effect of time in establishing causation hence our findings on factors associated with testing may require further research to account for the effect of time [41].

## Conclusion

Though majority of men had ever-tested for HIV, only a few had tested in the last 12-months despite having a continued risk of infection because of their lifestyles. This shows the need to emphasize regular testing for men. Factors associated with having ever-tested were largely similar to those of testing in the last 12 months. Men who were Catholic and aged 25–49 years with secondary or higher education were more likely to test. There is however a testing lag among youth (15–24 years) and older (50–83 years) men highlighting the need for further research to explore the influence of age on HIV testing needs among men. Subsequently, offering age-tailored HIV services could increase testing uptake among all men and reduce gender inequity in HTS.

## Supporting information

**S1 Table. HIV-related risky behaviors among male survey participants from selected villages in Mpigi district, August to September 2018.**
(DOCX)

**S1 File.**
(ZIP)

**S2 File.**
(ZIP)

## Acknowledgments

We extend special thanks to all study participants, the district health teams who enabled us to reach communities for data collection, the research assistants who ensured that we obtain good quality data, and the administrative teams who supported this study.

## Author Contributions

**Conceptualization:** Joanita Nangendo, Anne R. Katahoire, Gloria Odei Obeng-Amoako, Mercy Muwema, Charles A. Karamagi, Moses R. Kamya, Joan N. Kalyango.

**Formal analysis:** Mari Armstrong-Hough, Jane Kabami, Gloria Odei Obeng-Amoako, Charles A. Karamagi, Moses R. Kamya, Joan N. Kalyango.

**Funding acquisition:** Joanita Nangendo, Fred C. Semitala, Charles A. Karamagi, Moses R. Kamya.

**Investigation:** Joanita Nangendo, Mari Armstrong-Hough, Gloria Odei Obeng-Amoako, Joan N. Kalyango.

**Methodology:** Joanita Nangendo, Anne R. Katahoire, Mari Armstrong-Hough, Jane Kabami, Gloria Odei Obeng-Amoako, Mercy Muwema, Charles A. Karamagi, Rhoda K. Wanyenze, Moses R. Kamya, Joan N. Kalyango.

**Project administration:** Joanita Nangendo, Fred C. Semitala, Charles A. Karamagi, Moses R. Kamya, Joan N. Kalyango.

**Supervision:** Anne R. Katahoire, Charles A. Karamagi, Rhoda K. Wanyenze, Moses R. Kamya, Joan N. Kalyango.

**Validation:** Anne R. Katahoire, Jane Kabami, Gloria Odei Obeng-Amoako, Mercy Muwema.

**Visualization:** Rhoda K. Wanyenze, Moses R. Kamya.

**Writing – original draft:** Joanita Nangendo, Joan N. Kalyango.

**Writing – review & editing:** Joanita Nangendo, Anne R. Katahoire, Mari Armstrong-Hough, Jane Kabami, Gloria Odei Obeng-Amoako, Mercy Muwema, Fred C. Semitala, Charles A. Karamagi, Rhoda K. Wanyenze, Moses R. Kamya, Joan N. Kalyango.

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
