## [Decision Letter · Decision Letter 0]

14 May 2020

PONE-D-20-10107

Prevalence, associated factors and perspectives of HIV testing among men in Uganda

PLOS ONE

Dear Nangendo,

Thank you for submitting your manuscript to PLOS ONE. After careful consideration, we feel that it has merit but does not fully meet PLOS ONE’s publication criteria as it currently stands. Therefore, we invite you to submit a revised version of the manuscript that addresses the points raised during the review process.

We would appreciate receiving your revised manuscript by Jun 28 2020 11:59PM. To enhance the reproducibility of your results, we recommend that if applicable you deposit your laboratory protocols in protocols.io, where a protocol can be assigned its own identifier (DOI) such that it can be cited independently in the future. For instructions see: http://journals.plos.org/plosone/s/submission-guidelines#loc-laboratory-protocols

We look forward to receiving your revised manuscript.

Kind regards,

Joel Msafiri Francis, MD, MS, PhD

Academic Editor

PLOS ONE

Journal Requirements:

Reviewers' comments:

Reviewer's Responses to Questions

**Comments to the Author**

1. Is the manuscript technically sound, and do the data support the conclusions?

Reviewer #1: Yes

Reviewer #2: Yes

2. Has the statistical analysis been performed appropriately and rigorously? 

Reviewer #1: Yes

Reviewer #2: Yes

3. Have the authors made all data underlying the findings in their manuscript fully available?

Reviewer #1: Yes

Reviewer #2: Yes

4. Is the manuscript presented in an intelligible fashion and written in standard English?

Reviewer #1: Yes

Reviewer #2: Yes

5. Review Comments to the Author

Reviewer #1: Overall, this is a well written manuscript. The author has conducted a mixed methods study describing the prevalence and associated factors of HIV testing in an adult male population in two districts in Uganda, Mpigi and Kampala. The qualitative component described the barriers and facilitators of HIV testing in Ugandan men aged >15 years using focussed group discussions (FGDs)

However, there are some issues that need some further revision.

Major issues

1. The introdcution does not provide an adequate rationale for conducting this study. The current literature on prevalence and associated factors of HIV testing in men is not adequately described, in particular no mention is made of existing literature from Uganda. A more detailed literature review may be necessary as well as an adequate rationale for the current study.

2. The study objectives for the quantitative study state that the authors sought to measure the prevalence and associated factors of HIV testing in men> 15 years. However, in the results section starting from line 224-233, there is a a description of barriers, facilitators and views of how to improve HIV testing services for men measured as proportions.

It is unclear why this was described given that the qualitative componenent of the study also addresses this objective adequately. The quantitative description of these barriers and facilitators seems redundant in this instance. Please consider removing that section of the results and ammending the discussion accordingly. Alternatively, a justification should be given for including these resuls.

3. In the data analysis section, the authors state that a modified poisson regression analysis was used. However, the outcomes measured are binary, HIV testing in the last 12 months and life time testing. Furthermore, prevalence ratios are documented as the measures of association. Therefore, the analysis method is logistic regression and not Poisson regression is described in the manuscript.

Minor issues

1. In the study design section, the authors mention that the cross sectional survey was conducted in Mpigi distrcit only and not Kampala whislt the focus group discussions were conducted in both districts. Is there a justification for this?

2.In the section under sampling, the authors used the modified Kish Leslie formula to calculate the sample size. Can the equation for this formula be provided in the text?

3. In the section under sampling, an adequate description of the sampling interval is not provided.

4. In the sampling section, line 128, the authors reported using mobilizers to obtain their sample for the focus group discussions. However, it unclear how the selction criteria were applied by the mobilizers. An adequate description of how the study criteria were applied by the moblisers is needed. In addition, in line 131, more detail is needed to describe the process used to get to data saturation.

5. In the data collection section.the authors reprted that all FGDs were conducted in Luganda. Could all the men speak Luganda? The inclusion criteria stated that a participant was included if he could speak either English or Luganda. This needs to be made clearer.

6. In the results section, table 1, medians with minimum and maximum values are reported. Please report medians with IQRs as stated in the data analysis section.

7. In the results section, please report the numbers in addition to percentages when desscribing the prevalence of HIV testing.

8. This study excluded men who could not speak either Luganda or English. This may introduce selction bias, and this needs to be described as a limitation of the study. In addition, a descrption of the proportions of people who either speak English and/or Luganda would be useful to further understand the study context.

Reviewer #2: This is an interesting and important analysis of HIV testing among men in central Uganda. The paper is well presented, comprehensive, and clear. The mixed methods approach of this study is insightful lends a lot of depth to the data presented. I have a few minor comments meant for clarification.

1. Consider “HIV risk-related behavior” instead of “HIV risky behavior”.

2. Abstract, Conclusions, Line 44: consider “men’s past 12-month HIV test” or “recent” HIV testing instead of “testing at 12 months”.

3. For the composite variable related to HIV risk behavior, how was the variable operationalized? Was it engaging in any of the listed behaviors, or was it a score (number of behaviors)? It would be interesting to see the distribution of these behaviors, perhaps in a supplementary table?

4. Which variables were considered potentially confounding a priori and thus included in multivariable analyses?

5. I may be understanding the reporting of the interaction between marital status and risk behavior incorrectly, but I think standard interpretation of the interaction term for marital status would be effect of marital status when “HIV risk behavior==0”. So if HIV risk behavior was coded as 1=Yes, 0=No, then it would be that those who are widowed/divorced have reduced risk of HIV testing compared to those who are married if they do not engage in HIV risk behavior (risk==0)? Or was it reverse-coded?

6. For quantitative barriers to HIV testing, were participants asked to endorse a pre-populated list of potential barriers, or were this entered in some kind of free-form? If they were asked as a list of existing potential responses, how was the list of potential barriers decided?

7. Were data on HIV status available (even self-reported)? For men who hadn’t tested in the previous 12 months, was it possible they hadn’t tested because they already knew they were positive? If this information is not available, it may be worth noting in the discussion.

6. PLOS authors have the option to publish the peer review history of their article (what does this mean?). If published, this will include your full peer review and any attached files.

Reviewer #1: No

Reviewer #2: No

---

## [Author Response · Author response to Decision Letter 0]

16 Jun 2020

Reviewer #1: 

Major issues

1. The introduction does not provide an adequate rationale for conducting this study. The current literature on prevalence

and associated factors of HIV testing in men is not adequately described, in particular no mention is made of existing

literature from Uganda. A more detailed literature review may be necessary as well as an adequate rationale for the

current study.

Response: Thank you noting this. We have included some more literature on HIV testing and the associated factors among men in Uganda. See Line 57 – 66. 

2. The study objectives for the quantitative study state that the authors sought to measure the prevalence and associated

factors of HIV testing in men> 15 years. However, in the results section starting from line 224-233, there is a description

of barriers, facilitators and views of how to improve HIV testing services for men measured as proportions.

It is unclear why this was described given that the qualitative component of the study also addresses this objective

adequately. The quantitative description of these barriers and facilitators seems redundant in this instance. Please

consider removing that section of the results and amending the discussion accordingly. Alternatively, a justification

should be given for including these results.

Response: Thank you for raising this concern. We presented quantitative results on barriers, facilitators and views on how to improve HIV testing services to help us concretize the qualitative evidence on HIV testing perspectives which was collected in parallel. Since the survey was conducted only Mpigi district, the nature of results builds confidence about the generalizability of the findings throughout the study setting.

3. In the data analysis section, the authors state that a modified Poisson regression analysis was used. However, the outcomes measured are binary, HIV testing in the last 12 months and life time testing. Furthermore, prevalence ratios are documented as the measures of association. Therefore, the analysis method is logistic regression and not Poisson regression is described in the manuscript.

Response: Thank you for this comment. Yes, we used modified Poisson regression and not Logistic regression. The choice of using Poisson regression was because the outcome of interest (HIV testing in the last 12 months and life time testing was high with a prevalence of more than 5% hence using logistic regression would not be appropriate as relative risk would over-estimate odds of occurrence. However, since the outcome can be counted and was measured on a binary scale, prevalence ratios were a more resilient and preferred measure of association.

Minor issues

1. In the study design section, the authors mention that the cross sectional survey was conducted in Mpigi district only

and not Kampala whilst the focus group discussions were conducted in both districts. Is there a justification for this?

Response: Yes, conducted the survey on in Mpigi alone and the qualitative study in both districts. This is because Mpigi had a rather more stable population throughout its communities compared to Kampala which could ease sampling at the household. Also men in living in urban centers in both districts largely engaged in similar activities and lifestyles, hence the complementary qualitative study gave us opportunity to attain deeper understanding of their perspectives on HIV testing.

2. In the section under sampling, the authors used the modified Kish Leslie formula to calculate the sample size. Can the

equation for this formula be provided in the text?

Response: Thank you. The equation has been included. See Line 115 - 123.

3. In the section under sampling, an adequate description of the sampling interval is not provided.

Response: We have included some details. See Line 130 – 132.

4. In the sampling section, line 128, the authors reported using mobilizers to obtain their sample for the focus group

discussions. However, it is unclear how the selection criteria were applied by the mobilizers. An adequate description of how the study criteria were applied by the mobilisers is needed. In addition, in line 131, more detail is needed to describe the process used to get to data saturation.

Response: Thank you for identifying these. We have added the details of how the eligibility criteria was applied to the mobilizer and also elaborated on how data saturation was reached. See line 141 – 145.

5. In the data collection section. The authors repeated that all FGDs were conducted in Luganda. Could all the men speak

Luganda? The inclusion criteria stated that a participant was included if he could speak either English or Luganda. This

needs to be made clearer.

Response: Thank you for this comment. We have clarified this in the section about study participants. For the qualitative study, the eligibility criteria required that one could speak Luganda. See line 111 – 112.

6. In the results section, table 1, medians with minimum and maximum values are reported. Please report medians with

IQRs as stated in the data analysis section.

Response: We have replaced the minimum and maximum values with interquartile ranges as stated in the data analysis section. See Table 1.

7. In the results section, please report the numbers in addition to percentages when describing the prevalence of HIV

testing.

Response: Thank you, this has been included. See Line 230– 231.

8. This study excluded men who could not speak either Luganda or English. This may introduce selection bias, and this

needs to be described as a limitation of the study. In addition, a description of the proportions of people who either speak

English and/or Luganda would be useful to further understand the study context.

Response: Thank you, we have included this in the section on study limitations. See Line 522 – 524.

Reviewer #2: 

1. Consider “HIV risk-related behavior” instead of “HIV risky behavior”.

Response: We have replaced all “HIV risky behavior” to be “HIV risk-related behavior” throughout the manuscript.

2. Abstract, Conclusions, Line 44: consider “men’s past 12-month HIV test” or “recent” HIV testing instead of “testing at 12 months”.

Response: We have replaced the phrase in the conclusion of the abstract to read “Recent HIV testing among men remains low.” See line 44.

3. For the composite variable related to HIV risk behavior, how was the variable operationalized? Was it engaging in any

of the listed behaviors, or was it a score (number of behaviors)? It would be interesting to see the distribution of these

behaviors, perhaps in a supplementary table?

Response: Thank you for pointing this out. We created HIV risk behavior as a composite variable during analysis but at data collection, we collected data on individual variables (multiple sexual partners in the last 6 months, inconsistent condom use, engaging in transactional sex, use of alcohol and drugs, having sex after using drugs and/or alcohol). 

See table 5 in the supplement.

4. Which variables were considered potentially confounding a priori and thus included in multivariable analyses?

Response: The variable included in multivariable analysis for being considered as confounders were occupation and engaging in HIV risk-related behaviors. In our data, they were not significant at bivariate analysis for both levels of the outcome. See Table 3.

5. I may be understanding the reporting of the interaction between marital status and risk behavior incorrectly, but I think

standard interpretation of the interaction term for marital status would be effect of marital status when “HIV risk

behavior==0”. So if HIV risk behavior was coded as 1=Yes, 0=No, then it would be that those who are widowed/divorced

have reduced risk of HIV testing compared to those who are married if they do not engage in HIV risk behavior (risk==0)? Or was it reverse-coded?

Response: Thank you for this. It is true that HIV risk-related behavior was coded as 1=Yes and 0=No. However, from the figures that compared to the married, those who are widowed and divorced were less likely to test even when they engaged in risk-related behavior.

6. For quantitative barriers to HIV testing, were participants asked to endorse a pre-populated list of potential barriers, or

were this entered in some kind of free-form? If they were asked as a list of existing potential responses, how was the list

of potential barriers decided?

Response: The participants were asked to respond to a list of existing potential responses. This list was informed but findings from prior studies which used quantitative and qualitative approaches to study barriers to men’s HIV testing. These studies include: systematic reviews by Musheke and colleagues, 2013; Hensen and colleagues, 2014 as well as individual studies by DiCarlo et al., 2014; Camlin et al., 2016; Siu et al.., 2014; and Skovdal et al, 2011 among others.

7. Were data on HIV status available (even self-reported)? For men who hadn’t tested in the previous 12 months, was it

possible they hadn’t tested because they already knew they were positive? If this information is not available, it may be

worth noting in the discussion.

Response: Thank you for this insight. We have included it in the discussion. See Line 474 – 476.

---

## [Decision Letter · Decision Letter 1]

27 Jul 2020

Prevalence, associated factors and perspectives of HIV testing among men in Uganda

PONE-D-20-10107R1

Dear Dr. Nangendo,

We’re pleased to inform you that your manuscript has been judged scientifically suitable for publication and will be formally accepted for publication once it meets all outstanding technical requirements.

Kind regards,

Joel Msafiri Francis, MD, MS, PhD

Academic Editor

PLOS ONE

Additional Editor Comments (optional):

Please address the additional minor comment from reviewer 1 when proofreading the document for publication.

Reviewers' comments:

Reviewer's Responses to Questions

**Comments to the Author**

1. If the authors have adequately addressed your comments raised in a previous round of review and you feel that this manuscript is now acceptable for publication, you may indicate that here to bypass the “Comments to the Author” section, enter your conflict of interest statement in the “Confidential to Editor” section, and submit your "Accept" recommendation.

Reviewer #1: All comments have been addressed

Reviewer #2: All comments have been addressed

2. Is the manuscript technically sound, and do the data support the conclusions?

Reviewer #1: Yes

Reviewer #2: Yes

3. Has the statistical analysis been performed appropriately and rigorously? 

Reviewer #1: Yes

Reviewer #2: Yes

4. Have the authors made all data underlying the findings in their manuscript fully available?

Reviewer #1: Yes

Reviewer #2: Yes

5. Is the manuscript presented in an intelligible fashion and written in standard English?

Reviewer #1: Yes

Reviewer #2: Yes

6. Review Comments to the Author

Reviewer #1: Minor issue: Line 522 in the manuscript- add the word bias after the word selection. Tbe word should read selection bias, not selction.

Reviewer #2: (No Response)

7. PLOS authors have the option to publish the peer review history of their article (what does this mean?). If published, this will include your full peer review and any attached files.

Reviewer #1: No

Reviewer #2: No

---

## [Editor Report · Acceptance letter]

30 Jul 2020

PONE-D-20-10107R1 

Prevalence, associated factors and perspectives of HIV testing among men in Uganda 

Dear Dr. Nangendo:

I'm pleased to inform you that your manuscript has been deemed suitable for publication in PLOS ONE. Congratulations! Your manuscript is now with our production department. 

Kind regards, 

on behalf of

Dr. Joel Msafiri Francis 

Academic Editor

PLOS ONE